# Transcriptomic Analysis of the CAM Species *Kalanchoë fedtschenkoi* Under Low- and High-Temperature Regimes [note 1]

**DOI:** 10.3390/plants13233444

**Published:** 2024-12-08

**Authors:** Rongbin Hu, Jin Zhang, Sara Jawdy, Avinash Sreedasyam, Anna Lipzen, Mei Wang, Vivian Ng, Christopher Daum, Keykhosrow Keymanesh, Degao Liu, Alex Hu, Jin-Gui Chen, Gerald A. Tuskan, Jeremy Schmutz, Xiaohan Yang

**Affiliations:** 1Biosciences Division, Oak Ridge National Laboratory, Oak Ridge, TN 37831, USA; hu.rongbin@gmail.com (R.H.); zhangj@zafu.edu.cn (J.Z.); jawdys@ornl.gov (S.J.); degliu@ttu.edu (D.L.); chenj@ornl.gov (J.-G.C.); tuskanga@ornl.gov (G.A.T.); 2The Center for Bioenergy Innovation, Oak Ridge National Laboratory, Oak Ridge, TN 37831, USA; 3State Key Laboratory of Subtropical Silviculture, School of Forestry and Biotechnology, Zhejiang A&F University, Hangzhou 311300, China; 4HudsonAlpha Institute for Biotechnology, 601 Genome Way, Huntsville, AL 35801, USA; asreedasyam@hudsonalpha.org (A.S.); jschmutz@hudsonalpha.org (J.S.); 5Department of Energy Joint Genome Institute, Lawrence Berkeley National Laboratory, Berkeley, CA 94589, USA; alipzen@lbl.gov (A.L.); mwang@lbl.gov (M.W.); vng@lbl.gov (V.N.); cgdaum@lbl.gov (C.D.); kkeymanesh@lbl.gov (K.K.); 6Institute of Genomics for Crop Abiotic Stress Tolerance, Department of Plant and Soil Science, Texas Tech University, Lubbock, TX 79409, USA; 7Department of Chemical and Environmental Engineering, University of California-Riverside, Riverside, CA 92521, USA; flyingalex.hu@gmail.com

**Keywords:** crassulacean acid metabolism, RNA-Seq, gene regulation, heat shock proteins

## Abstract

Temperature stress is one of the major limiting environmental factors that negatively impact global crop yields. *Kalanchoë fedtschenkoi* is an obligate crassulacean acid metabolism (CAM) plant species, exhibiting much higher water-use efficiency and tolerance to drought and heat stresses than C_3_ or C_4_ plant species. Previous studies on gene expression responses to low- or high-temperature stress have been focused on C_3_ and C_4_ plants. There is a lack of information about the regulation of gene expression by low and high temperatures in CAM plants. To address this knowledge gap, we performed transcriptome sequencing (RNA-Seq) of leaf and root tissues of *K. fedtschenkoi* under cold (8 °C), normal (25 °C), and heat (37 °C) conditions at dawn (i.e., 2 h before the light period) and dusk (i.e., 2 h before the dark period). Our analysis revealed differentially expressed genes (DEGs) under cold or heat treatment in comparison to normal conditions in leaf or root tissue at each of the two time points. In particular, DEGs exhibiting either the same or opposite direction of expression change (either up-regulated or down-regulated) under cold and heat treatments were identified. In addition, we analyzed gene co-expression modules regulated by cold or heat treatment, and we performed in-depth analyses of expression regulation by temperature stresses for selected gene categories, including CAM-related genes, genes encoding heat shock factors and heat shock proteins, circadian rhythm genes, and stomatal movement genes. Our study highlights both the common and distinct molecular strategies employed by CAM and C_3_/C_4_ plants in adapting to extreme temperatures, providing new insights into the molecular mechanisms underlying temperature stress responses in CAM species.

## 1. Introduction

Global climate change has increasingly subjected plants to extreme weather conditions such as heat and cold stresses, leading to significant yield reductions in major crops such as cotton, maize, wheat, rice, and soybean [1,2,3,4]. In general, an increase in temperature by 10–15 °C above ambient conditions is considered heat stress, which disrupts cellular functions, reduces photosynthetic efficiency, and accelerates water loss through transpiration [5,6]. Conversely, exposure to low temperatures (0–15 °C) that do not reach freezing point can cause cold stress, leading to inhibited enzymatic activity, damage to cellular structures, and impaired nutrient uptake [7,8]. In response, plants have evolved a range of adaptive strategies and regulatory mechanisms to mitigate temperature stresses. For instance, three genes of the *C-REPEAT BINDING FACTOR/DEHYDRATION RESPONSIVE ELEMENT (DRE)- BINDING1* (*CBF*/*DREB1*) family function as pivotal transcription factors that initiate the expression of *COLD-RESPONSIVE* (*COR*) genes, thereby enhancing cold tolerance in *Arabidopsis* [9,10,11,12]. On the other hand, under heat stress, the *HEAT SHOCK FACTOR A1* (*HSFA1*) gene acts as a critical transcription factor, regulating the expression of heat-responsive genes [13,14,15]. Moreover, HEAT SHOCK PROTEIN 70 (HSP70) and HSP90 have been identified as negative regulators of HSFA1-mediated heat stress responses in *Arabidopsis* and tomato [16,17,18,19,20].

Plants performing Crassulacean acid metabolism (CAM) photosynthesis exhibit much higher water-use efficiency (WUE) and tolerance to heat and drought stresses than plants performing C_3_ and C_4_ photosynthesis, and consequently natural or engineered CAM plants (e.g., C_3_ plants engineered with CAM genes) have great potential for sustainable crop production in arid and semi-arid regions [21,22,23,24,25]. During the nighttime, CAM plants open stomata to fix CO_2_, via the carboxylation process, into malate, which is then converted into malic acid and stored in the vacuole. During the daytime, malic acid is exported out of the vacuole and decarboxylated to release CO_2_ for refixation through the Calvin cycle while stomata remain closed to minimize water loss [21,23,26,27,28,29]. *Kalanchoë fedtschenkoi* is a native species of Madagascar that performs obligate CAM photosynthesis. It is characterized by its relatively small size, ease of laboratory maintenance, preference for dry soil and bright, indirect light [30]. With its extensive genomic resources, this species has emerged as an excellent model for in-depth studies on the molecular mechanism underlying CAM physiology, facilitating a better understanding of how CAM species have evolved in adaptation to arid environments [27,28,29,31,32,33,34]. In particular, recent transcriptomic analyses have provided new insights into the gene regulation and molecular responses in *K. fedtschenkoi* under various light intensities, light qualities, and drought stress conditions [28,29,34]. 

CAM plants thrive in desert environment and exhibit superior acclimation to both warm and cold temperatures compared to C_3_ and C_4_ plants due to the large temperature difference between day and night in deserts. Species of *Opuntia* are particularly noted for their ability to withstand extreme heat and cold stresses. For instance, *O. robusta* can tolerate high temperature up to 61 °C and low temperature down to −8 °C during one hour treatment [35]. The Rubisco activase (RCA) is a key enzyme engaged in activation of Rubisco, and its state is thermally sensitive [36,37]. Notably, RCA isoforms from a CAM species *Agave tequilana* display 10 °C higher thermostability than those from the C_3_ plant rice (*Oryza sativa*) [38]. Additionally, it was reported that *HEAT SHOCK PROTEINS* (*HSPs*) showed convergent changes in diel transcript expression in CAM species, highlighting their importance in responding to heat stress [27]. 

Many transcriptomic datasets have been published on the cold and heat stress responses in C_3_ and C_4_ species, providing valuable insights into the gene regulatory mechanisms underlying their adaptation to temperature stresses [39,40,41,42]. However, there has been a noticeable lack of studies on the gene regulation in response to cold and heat stresses in CAM species, leaving a significant gap in our understanding of how these plants cope with temperature stress. To address this knowledge gap, in this study, we generated a comprehensive RNA-seq dataset using mature leaves and roots of *K. fedtschenkoi* under cold and heat stress conditions at two time points: dawn (2 h before the start of the light period) and dusk (2 h before the start of the dark period) [34]. Using this RNA-seq dataset, we identified differentially expressed genes (DEGs) under these stress conditions and constructed a weighted gene co-expression network to identify gene modules responsive to different temperatures stresses. Furthermore, we performed detailed analysis of expression response to low and high temperatures for genes relevant to CAM physiology and temperature stress tolerance.

## 2. Results

### 2.1. Quality of Transcriptome-Sequencing (RNA-Seq) Data

To evaluate the quality of the RNA-Seq data, we initially conducted a principal component analysis and identified principal component 1 (PC1) and PC2 as accounting for 29.06% and 20.94% of the variance in the gene expression dataset, respectively (Appendix A). The three biological replicates for different treatments (cold, control, and heat) and two types of tissues (leaf and root) were closely clustered, whereas the data from two time points (dawn and dusk) were well separated (Appendix A), implying the high reliability and reproducibility of our dataset.

### 2.2. Transcriptomic Response to Low Temperature in the Leaf Tissues of K. fedtschenkoi

Comparative analysis of gene expression in the leaf tissue between cold treatment and control was conducted to identify DEGs responsive to cold stress at dusk and dawn (Appendix A). At dusk, a total of 4087 DEGs were identified, with 1891 down-regulated and 4392 up-regulated, and at dawn a total of 6291 DEGs, of which 3510 and 2781 genes were down- and up-regulated, respectively, between the two conditions (Figure 1a; Table 1). Notably, the DEG lists include several *K. fedtschenkoi* genes that have orthologous genes reported to be regulated by low temperature in other species. For example, two *RARE-COLD-INDUCIBLE 2B* (*RCI2B*) genes (Kaladp0024s0843 and Kaladp0057s0094) and one *TEMPERATURE-INDUCED LIPOCALIN* (*TIL*) gene (Kaladp0092s0126) were up-regulated in response to low temperatures, while a *C-repeat-binding factor 4* (*CBF4*) gene (Kaladp0046s0337) and a *COLD, CIRCADIAN RHYTHM*, and *RNA BINDING 2* (*CCR2*) gene (Kaladp0020s0114) were down-regulated (Appendix A). The orthologous genes of *RCI2B*, *TIL*, *CBF4*, and *CCR2* in other plant species (e.g., *Arabidopsis*, wheat and *Medicago falcata*) showed cold-regulated expression similar to that in *K. fedtschenkoi* [43,44,45,46,47]. At dawn, we identified a total of 6291 DEGs, of which 3510 and 2781 genes were down- and up-regulated under cold- or heat-stress conditions in leaf tissue, respectively (Figure 1a; Table 1). Interestingly, the *RCI2B* gene (Kaladp0024s0843) was up-regulated while the *CBF4* gene (Kaladp0046s0337) was down-regulated by cold treatment at dawn, showing expression response similar to that at dusk. In addition, two *COLD REGULATED GENE* 27 (*COR27*) genes (Kaladp0042s0067 and Kaladp0089s0010), one *CCR1* gene (Kaladp0018s0148) and two *CCR2* genes (Kaladp0020s0114 and Kaladp0033s0217) were up-regulated by low temperature treatment. In contrast, the expression of a *COLD REGULATED 314 THYLAKOID MEMBRANE 2* (*COR314-TM2*) gene (Kaladp0021s0083), an *RCI2A* gene (Kaladp0053s0168) and another *TIL* gene (Kaladp0060s0055) were down-regulated by cold stress (Appendix A).

We performed gene ontology (GO)-enrichment analysis for these DEG sets (Figure 1c; Appendix A). In the category biological process (BP), the up-regulated genes under cold stress condition were enriched in “phosphorus metabolic process” and “phosphate metabolic process” at dawn and enriched in “biosynthetic process” and “cellular biosynthetic process” at dusk. The down-regulated genes in the roots under cold stress conditions were enriched in “biological regulation”, “regulation of biological process”, and “regulation of cellular process” at dawn, and the “carbohydrate metabolic process” was enriched for down-regulated genes at dusk (Appendix A).

### 2.3. Transcriptomic Response to Low Temperature in the Root Tissues of K. fedtschenkoi

Compared to leaf tissues, a smaller number of genes were differentially expressed under cold stress in root tissues. Specifically, we identified 1021 and 1953 DEGs that were down-regulated, while 847 and 830 genes were up-regulated by cold stress at dawn and dusk, respectively (Figure 1b; Table 1). Notably, at dusk, the expression of a *COR27* gene (Kaladp0011s1228) and a *Heat Shock Factor 4* (*HSF4*) gene (Kaladp0028s0067) was up-regulated, whereas two *HSF* genes (Kaladp0011s0272 and Kaladp0015s0018) were down-regulated in response to low temperature. At dawn, these two *HSF* genes (Kaladp0011s0272 and Kaladp0015s0018) remained down-regulated at dusk, while three other *HSF* genes (Kaladp0010s0067, Kaladp0012s0036, and Kaladp0102s0147) were up-regulated under cold stress (Appendix A).

The up-regulation of *COR27*, a gene associated with cold acclimation, in both leaf and root tissues implies the importance of this gene in enhancing cold stress tolerance across the whole plant. In addition, HSFs are typically known for their roles in the heat stress response [1,48,49,50], but their involvement in cold stress in both *K. fedtschenkoi* (as revealed in this study) and non-CAM species stress highlights the functional diversification of the HSF family in response to temperature stresses. The differential expression of specific HSFs indicates an adaptive response aimed at stabilizing protein structures and preventing cold-induced damage in root tissues.

GO-enrichment analysis revealed that up-regulated DEGs in the root under the cold condition were highly enriched in “biosynthetic process” at dawn and in “transmembrane transport” at dusk. For down-regulated DEGs, the GO terms were enriched in “regulation of metabolic process” and “regulation of gene expression” at dawn and enriched in “oxidation reduction” and “regulation of cellular process” at dusk (Appendix A).

### 2.4. Transcriptomic Response to High Temperature in the Leaf Tissues of K. fedtschenkoi

For the heat stress treatment, we identified a total of 9089 DEGs in leaf tissues at dawn, with 4774 and 4315 genes down- and up-regulated, respectively. At dusk, 8806 DEGs were identified, including 4640 down-regulated and 4166 up-regulated genes (Figure 2a; Table 1). Among these DEGs, genes encoding HSFs and HSPs were over-represented (*p* ≤ 0.05), reflecting their critical roles in the heat stress response. Specifically, at dusk, multiple *HSF* genes showed up-regulation, including *HSFA2* (Kaladp0081s0271, Kaladp0102s0147 and Kaladp0515s0230) and *HSFA3* (Kaladp0011s0272), which are known to be key regulators of the heat stress response, driving the expression of various HSPs in plants [49,50]. Several HSP genes, such as *HSP70* (Kaladp0039s0620) and *HSP90* (Kaladp0001s0054 and Kaladp0024s0163), were also significantly up-regulated, consistent with its role in protein folding and protection against heat-induced denaturation [1,50]. Interestingly, at dawn, the three *HSFA2* genes (Kaladp0081s0271, Kaladp0102s0147, and Kaladp0515s0230) and two *HSP90* genes (Kaladp0001s0054 and Kaladp0024s0163) remained up-regulated by heat stress as at dusk, indicating sustained activation of the heat stress response across different times of day (Appendix A). The persistent up-regulation of these *HSF* and *HSP* genes in *K. fedtschenkoi* under heat stress suggests that CAM plants are maintaining a heightened state of readiness to counteract the damaging effects of elevated temperatures similar to those mechanisms known in C_3_ and C_4_ plants.

In addition to *HSF*s and *HSP*s, we found that other genes involved in oxidative stress defense were differentially expressed under heat stress conditions in comparison to the control condition. Notably, two *ASCORBATE PEROXIDASE 2* (*APX2*) genes (Kaladp0610s0002 and Kaladp0058s0448) were up-regulated by heat stress at both dusk and dawn. One *APX1* gene (Kaladp0071s0237) and another *APX2* gene (Kaladp0610s0002) were up-regulated by heat stress at dawn only. In terms of catalase activity, a *CATALASE 1* (*CAT1*) gene (Kaladp0108s0002) and a *CAT2* gene (Kaladp0052s0025) were up-regulated at dusk, whereas another *CAT1* gene (Kaladp0114s0009) and a different *CAT2* gene (Kaladp0001s0016) were up-regulated at dawn (Appendix A). These findings suggest the multifaceted nature of the plant’s response to heat stress, which not only involves chaperone-mediated protein protection but also enhances the scavenging of reactive oxygen species (ROS) generated under high-temperature conditions.

Our GO-enrichment analysis of DEGs in leaf tissues under heat stress revealed that the up-regulated genes were enriched in “primary metabolic process” and “cellular metabolic process” at dawn and in “response to stress” at dusk. For down-regulated genes at dawn or dusk under heat conditions, the GO terms were enriched in “establishment of localization” and “transport”, respectively (Figure 1c; Appendix A).

### 2.5. Transcriptomic Response to High Temperature in the Root Tissues of K. fedtschenkoi

DEGs were also identified in root tissues in response to heat stress. We observed that 1810 and 2003 genes were up-regulated, while 2641 and 3327 genes were down-regulated at dawn and dusk, respectively (Figure 1b; Table 1). Notably, three *HSFA2* genes (Kaladp0081s0271, Kaladp0102s0147, and Kaladp0515s0230), one *HSFA3* gene (Kaladp0011s0272), and one *HSP70* gene (Kaladp0039s0620) were consistently up-regulated at both dusk and dawn, highlighting the critical roles that HSFs and HSPs play in protecting plants against high-temperature stress. Also, two *APX2* genes (Kaladp0610s0002 and Kaladp0058s0448) showed consistent up-regulation under heat treatment of the root at dusk and dawn. In addition, a *SUPEROXIDE DISMUTASE 2* (*SOD2*) gene (Kaladp0045s0159) was up-regulated at both dusk and dawn (Appendix A).

GO-enrichment analysis of these DEGs revealed that the up-regulated genes were enriched in “biological regulation” and “regulation of biological process” at dawn, and the GO term was enriched in “protein folding” at dusk. In contrast, the down-regulated genes were highly enriched in “oxidation reduction” at both dawn and dusk (Figure 1c; Appendix A).

### 2.6. Concordant Gene Regulation by Cold and Heat Stress in K. fedtschenkoi

To investigate the regulation of genes by cold and heat stress across different tissues and times of day, a Venn diagram analysis was performed using the identified DEGs (Figure 2a,b). In leaf tissues at dawn, a total of 3429 DEGs showed concordant regulation by cold and heat treatments (i.e., a gene shows the same direction of expression change (either up-regulated or down-regulated) under both treatments), with 1209 genes up-regulated and 1727 genes down-regulated (Figure 2a). At dusk, 2038 DEGs were concordantly regulated by both cold and heat treatments, including 667 up-regulated and 676 down-regulated genes (Figure 2a). Notably, three *HSF* genes (Kaladp0081s0271, Kaladp0102s0147, and Kaladp0515s0230) were up-regulated under both cold and heat stress conditions at dusk and dawn. Additionally, at dusk, another *HSF* gene (Kaladp0010s0067), a *CAT2* gene (*Kaladp0052s0025*), and a *TIL* gene (*Kaladp0092s0126*) were up-regulated by both cold and heat stress. At dawn, two *COR27* genes (Kaladp0011s1228 and Kaladp0089s0010), a *CCR1* gene (Kaladp0018s0148), two *CCR2* genes (Kaladp0020s0114), and one *HSF* gene (Kaladp0060s0167) were up-regulated in response to both cold and heat stress (Appendix A). The significant overlap of DEGs under cold and heat stress, including both up-regulated and down-regulated genes, suggests that certain pathways and regulatory networks are activated in response to temperature changes, regardless of whether the stress is caused by cold or heat. This indicates a degree of crosstalk between the cold and heat stress responses.

In root tissues at dawn, 655 DEGs were concordantly regulated by cold and heat treatments, with 62 up-regulated and 72 down-regulated genes (Figure 2b). At dusk, 1132 DEGs were common to both cold and heat stress, with 74 genes up-regulated and 690 genes down-regulated (Figure 2b). Specifically, a *COR27* gene (Kaladp0011s1228) was up-regulated by both cold and heat stress at dusk. The expression of two *HSF* genes (Kaladp0012s0036 and Kaladp0102s0147) was up-regulated under both cold and heat stress at dawn (Appendix A). These findings reveal tissue-specific differences in the stress response, with less genes concordantly regulated by cold and heat treatments in roots compared to leaves.

Our GO-enrichment analysis of the shared DEGs revealed that in leaf tissues, up-regulated genes were highly enriched in “macromolecule metabolic process” and “cellular macromolecule metabolic process” at dawn, whereas no enrichment was detected at dusk. The shared down-regulated DEGs were enriched in “photosynthesis” and “ion transport” at dawn and in “oxidation reduction” and “carbohydrate metabolic process” at dusk (Figure 2c, Appendix A). In root tissues, we found that the shared up-regulated DEGs were enriched in “polyol biosynthetic process” and “inositol biosynthetic process” at both dawn and dusk. At dusk, the GO terms were highly enriched in “oxidation reduction” and “response to stimulus” (Figure 2c, Appendix A).

### 2.7. Opposite Gene Regulation by Cold and Heat Stress in K. fedtschenkoi

Besides the genes concordantly regulated by cold and heat treatments, we also identified DEGs exhibiting the opposite regulation under cold and heat stress conditions (i.e., a gene shows opposite directions of expression change under cold and heat treatments). In leaf tissues, a total of 493 genes at dawn and 695 genes at dusk showed opposite directions of expression change in response to cold and heat treatments. Of these, 157 genes were oppositely regulated at both time points (Figure 3a). For instance, five MYB domain transcription factors (Kaladp0003s0134, Kaladp0011s0430, Kaladp0016s0076, Kaladp0045s0314, and Kaladp0075s0039) were up-regulated in response to cold stress but down-regulated under heat stress at both dawn and dusk (Figure 3c; Appendix A).

In root tissues, a similar trend of opposite gene regulation was observed. We identified 521 genes at dawn and 368 genes at dusk showing opposite regulation under cold and heat stress, among which 204 genes were shared at dawn and dusk (Figure 3b). Notably, a *HEAT SHOCK FACTOR A3* (*HSFA3*, Kaladp0011s0272) gene was down-regulated under cold stress but up-regulated under heat stress at both dawn and dusk (Appendix A). In addition, four MYB domain transcription factors (Kaladp0008s0500, Kaladp0011s0430, Kaladp0055s0531, and Kaladp0066s0109) were down-regulated in response to cold stress and up-regulated in response to heat stress at both time points (Figure 3d; Appendix A).

### 2.8. Gene Co-Expression Modules Regulated by Temperature Stresses

To uncover potential gene co-expression modules regulated by temperature stresses in K. fedtschenkoi, we performed a weighted gene co-expression network analysis (WGCNA). As shown in Figure 4, we characterized a total of 12 distinct co-expression modules, each represented by different color labels. The genes within the “black” and “magenta” modules were up-regulated by cold stress but down-regulated by heat treatment in leaf and root samples at dawn and dusk. The GO terms of “small molecule metabolic process” and “lipid metabolic process” were enriched for the “black” module. Genes in the “magenta” module were functionally enriched in “transmembrane transport” and “ion transport” (Appendix A). The DEGs in the “brown” module were up-regulated under heat stress condition in comparison to that of normal conditions at dawn and dusk in leaf and root tissues, but they were down-regulated under cold stress in root samples. This module was functionally enriched in “biological regulation”, macromolecule modification”, and “protein modification process”. Expression induction was also observed in the “yellow” module under heat stress but not under cold conditions at the dawn and dusk time points in leaf and root samples, which was functionally enriched in “primary metabolic process” and “cellular metabolic process” (Appendix A). Additionally, we identified transcription factors associated with various co-expression modules (Appendix A).

### 2.9. Responses of CAM Genes to Temperature Stresses

The CAM carboxylation process involves multiple genes, including *BETA-CARBONIC ANHYDRASE* (*β-CA*), *PHOSPHOENOLPYRUVATE CARBOXYLASE* (*PEPC*), *PHOSPHOENOLPYRUVATE CARBOXYLASE KINASE* (*PPCK*), *NAD(P)-MALATE DEHYDROGENASE* (*MDH*), and *ALUMINUM-ACTIVATED MALATE TRANSPORTER* (*ALMT*) genes [27]. Under cold or heat stress conditions, there was a notable reduction in the expression of two *β-CA* genes (Kaladp0024s0122 and Kaladp0034s0051) and *PPCK1* gene (Kaladp0037s0517) at dawn. The expression of *MDH* (Kaladp0082s0194) and *ALMT* (kaladp0073s0021) genes was down-regulated under cold or heat stress conditions at dusk (Figure 5). During the CAM decarboxylation process, cold or heat stress led to the down-regulation of two *PYRUVATE PHOSPHATE DIKINASE* (*PPDK*) genes (Kaladp0039s0092 and Kaladp0076s0229) and two *NAD-MALIC ENZYME* (*NAD-ME*) genes (Kaladp0033s0124 and Kaladp0472s0027) at both dawn and dusk. The *TONOPLAST DICARBOXYLATE TRANSPORTER* (*TDT*) gene (Kaladp0042s0251) was up-regulated under cold conditions at dusk but down-regulated under both cold and heat stress at dawn (Figure 5). These findings indicate that the core CAM regulatory genes are highly sensitive to temperature stress, with the majority being down-regulated in response to both cold and heat stress treatments, suggesting a disruption in the plant’s ability to efficiently perform crassulacean acid metabolism.

### 2.10. Responses of Heat Shock Factors and Proteins to Temperature Stresses

Plant heat shock factors (HSFs) and heat shock proteins (HSPs) play crucial roles in regulating responses to high- and low-temperature stresses [1,48,49,50]. Our analysis of differential gene expression between temperature stress treatment and the control identified 15 *HSF* genes affected by cold stress and 15 by heat stress in leaf tissues (Appendix A). Notably, seven HSF genes were regulated by both cold and heat stress treatments at dawn and dusk (Figure 6a). For instance, an *HSFA2* gene (Kaladp0081s0271) and an *HSFB1* gene (Kaladp0060s0167) were up-regulated by heat stress at dawn and dusk and also up-regulated under cold stress conditions at dawn. An *HSFC1* gene (Kaladp0031s0107) was down-regulated at both dawn and dusk under cold stress conditions, whereas it was up-regulated by heat stress treatment at dawn. Another gene, *HSFA1* (Kaladp0010s0067), showed increased transcript levels in response to heat stress at dawn and dusk and to cold stress at dusk (Figure 6a).

In root tissues, a total of seven *HSF* genes were identified to be responsive to cold stress, while eleven *HSF* genes were found to be affected by heat stress (Appendix A). Notably, there were five *HSF* genes that were regulated by both cold and heat stress treatments (Figure 6b). In particular, an *HSFA3* gene (Kaladp0011s0272) exhibited opposite directions of expression change under cold and heat treatments: it was down-regulated by cold stress but up-regulated by heat stress at both dawn and dusk. An *HSFB1* gene (Kaladp0028s0067) gene showed increased expression in response to cold stress specifically at dusk, whereas its transcript levels were reduced during heat stress at both dawn and dusk. Furthermore, an *HSFA2* gene (Kaladp0102s0147) was up-regulated under cold stress at dawn and also exhibited elevated transcript levels under heat stress at both dawn and dusk (Figure 6b).

Additionally, we examined the differentially expressed *HSP*-encoding genes under cold and heat stress conditions at dawn and dusk in both leaf and root tissues (Appendix A). In leaf tissues, a total of 26 *HSP* genes were affected by cold stress, while 38 *HSP* genes were impacted by heat stress. Among these, 12 *HSP* genes were concordantly regulated by cold and heat treatments at dawn and dusk (Appendix A). In root tissues, we identified 20 *HSP* genes that responded to cold stress and 51 to heat stress. Sixteen *HSP* genes were regulated by both cold and heat stress at dawn and dusk (Appendix A). Notably, three *HSP* genes (Kaladp0048s0389, Kaladp0053s0650, and Kaladp0051s0089) were found to be concordantly regulated by low and high temperatures in both leaf and root tissues.

### 2.11. Responses of Circadian Rhythm Gene and Stomatal Movement Genes to Temperature Stresses

The circadian clock acts as a molecular device to precisely control gene expression and promote stress signaling response and metabolic reprogramming [51,52]. We investigated the transcript profiles of circadian clock-related genes under low- and high-temperature stresses based on previous studies (Appendix A and Appendix A). Notably, the expression of *CONSTITUTIVE PHOTOMORPHOGENIC 1* (*COP1*) gene (Kaladp0011s0927) was induced by heat stress at dusk in leaf tissues. The *GIGANTEA* (*GI*) gene (Kaladp0040s0489) was up-regulated in response to low and high temperatures at dusk. Conversely, the *CIRCADIAN CLOCK ASSOCIATED 1* (*CCA1*) gene (Kaladp0496s0018) was significantly down-regulated by cold and heat stress at dawn. Two *REVEILLE* genes (*RVE1*: Kaladp0574s0015; *RVE8*: Kaladp0577s0020) were down-regulated under cold and heat stress conditions at dawn.

In addition, we examined the expression of genes involved in stomatal movement under temperature stress (Appendix A and Appendix A). *ABSCISIC ACID (ABA) INSENSITIVE 1* (*ABI1*: Kaladp0011s0443) was down-regulated at dawn in response to cold and heat stress. A *CALCIUM-DEPENDENT PROTEIN KINASE 3* (*CPK3*) gene (Kaladp0042s0341) and a *CPK6* gene (Kaladp0055s0096) were up-regulated by cold and heat stress at dawn and dusk, though with different levels of expression change. These results indicate that both circadian clock genes and stomatal movement genes were responsive to low- and high-temperature stress.

## 3. Discussion

Temperature stresses, including cold and heat stresses, significantly impact plant growth and development, often leading to reduced crop yields and subsequently affecting food security and agricultural sustainability [53,54,55]. To investigate gene expression regulation by temperature stresses in the CAM species *K. fedtschenkoi*, we generated RNA-Seq data using the leaf and root tissues collected at two different time points (dusk and dawn) from the plants grown under low (8 °C), normal (25 °C), and high (37 °C) growth conditions (Appendix A). These RNA-Seq data allowed us to comprehensively analyze the gene expression regulation by cold and heat treatments in CAM species.

DEGs were identified through comparative analysis between cold/heat stress treatment and control (normal condition) (Figure 1a,b). The *CBF-COR* pathway is well established as a critical regulatory mechanism or cold stress response in *Arabidopsis thaliana*. The CBF transcription factors, particularly *CBF1*, *CBF2*, and *CBF3*, activate the expression of COR genes, which help plants to acclimate to temperatures by enhancing cold tolerance mechanisms such as cryoprotection and stabilization of cellular structures [44,45]. Similarly, our results indicated that several DEGs in *K. fedtschenkoi*, including *CBF4*, *COR27*, and *COR314-TM2*, play important roles in cold stress regulation (Appendix A). The involvement of these genes in cold-responsive signaling in both *A. thaliana* and *K. fedtschenkoi* suggests a conserved cold-response mechanism between C_3_ and CAM species. In addition, our analysis identified several additional genes that were regulated by cold stress in *K. fedtschenkoi*, including *CCR1* and *CCR2* and *RCI2A* and *RCI2B* (Appendix A). These genes have also been reported to play significant roles in other plant species (e.g., *Arabidopsis*, wheat, and *Medicago falcata*) in response to cold temperatures, contributing to various protective mechanisms [43,44,46,47,56,57]. Our findings on the regulation of these genes highlight both shared and species-specific cold-responsive pathways compared to other plant species and emphasize the complexity and specificity of the cold stress adaptation at different times of the day in *K. fedtschenkoi*, suggesting that multiple pathways and regulatory networks are involved in ensuring plant survival under low-temperature conditions.

Heat stress triggers a complex response in plants, involving a variety of genes and regulators that contribute to protection and adaptation under high-temperature conditions. Heat shock factors (HSFs), particularly HSFAs (e.g., HSFA2, HSFA3), act as transcriptional regulators that activate the expression of HSPs when plants are exposed to elevated temperatures [49,50]. In C_3_ and C_4_ plant species HSPs function as molecular chaperones, preventing the aggregation of denatured proteins and facilitating the refolding of damaged proteins, which is crucial for maintaining cellular homeostasis under heat stress [50,58]. In this study, we observed up-regulation of *HSFA2* and *HSFA3* under heat stress, indicating their critical role in modulating the expression of HSPs for enhancement of heat tolerance (Appendix A). Expression induction of *HSP70* and *HSP90* under heat stress suggests their important function in signaling activation and protein stabilization. Furthermore, high-temperature stress results in reactive oxygen species (ROS) accumulation, thereby leading to oxidative damage [59]. The up-regulation of *APX1/2*, *SOD2* and *CAT* genes under heat stress in *K. fedtschenkoi* indicates an enhanced antioxidant defense mechanism aimed at detoxifying ROS and protecting cells from oxidative damage. Overall, our findings suggest the multifaceted nature of the plant’s response to heat stress, which not only involves chaperone-mediated protein protection but also enhances the scavenging of reactive oxygen species (ROS) generated under high temperature conditions.

In addition to their well-known role in heat stress, HSFs and HSPs are also activated in response to low temperature stress [1,48,49,50]. For instance, out of the 23 *OsHSF* genes in Oryza sativa, 16 were up-regulated after 10 minutes of exposure to heat stress, while 10 *OsHSF* genes were induced in response to low-temperature stress [60]. In *Arabidopsis*, in response to low temperature stress, NON-EXPRESSER OF PATHOGENESIS-RELATED GENES 1 (NPR1) interacts with HSFA1 to promote downstream gene expression and activate signaling pathways essential for cold acclimation [61]. Transcript accumulation of *HSP17.4CI*, *HSP70* and *HSP90* was observed in *Arabidopsis* after exposure to low temperature conditions [62,63]. In *K. fedtschenkoi*, our analysis revealed that multiple *HSFs* were responsive to either cold or heat stress in leaf and root samples (Appendix A). Notably, seven genes in the leaf and five genes in the root were regulated under both low and high temperature stress conditions (Figure 6). For example, the *HSFA3* gene (Kaladp0011s0272) was down-regulated under cold stress but up-regulated in response to heat stress in leaf tissues. Conversely, the *HSFB1* gene (Kaladp0028s0067) exhibited an opposite pattern, being up-regulated under cold conditions and down-regulated under heat conditions, suggesting a stress-specific regulation mechanism. Furthermore, a number of *HSPs* were identified as differentially expressed in response to low or high temperature stress (Appendix A). These findings provide new insights into the distinct molecular regulation of HSFs and HSPs in CAM species, and highlight the unique temperature stress adaptation strategies in CAM plants and expands our understanding of how these regulatory proteins contribute to the resilience of CAM species under environmental stress. 

In this study, we identified some *K. fedtschenkoi* genes that were concordantly regulated by both cold and heat stress in leaf and root tissues (Figure 2), indicating a degree of crosstalk between the cold and heat stress responses, allowing plants to deploy a coordinated defense mechanism. These findings suggest that these genes are good candidates for genetic engineering to improve tolerance to both cold and heat stresses. On the other hand, a number of DEGs were oppositely regulated by cold and heat stress at both dawn and dusk (Figure 3), suggesting that some genes can play opposite roles in response to cold and heat stresses in CAM plants. Furthermore, the observed opposite regulation of genes between cold and heat stress conditions suggests that *K. fedtschenkoi* employs distinct but interconnected signaling pathways to manage these two types of temperature stress and optimize their survival and growth across a wide range of environmental temperatures.

Our previous RNA-Seq analysis revealed that *K. fedtschenkoi* genes encoding core enzymes involved in carboxylation and decarboxylation processes were not significantly affected by drought stress [29]. However, in this study, we found that these CAM genes, including *β-CA*, *PPCK*, *MDH*, *ALMT*, *PPDK*, and *NAD-ME*, were mostly down-regulated by low- or high-temperature stress at dawn and dusk (Figure 5). Similarly, it was recently reported that heat stress significantly down-regulated transcript profiles of core C_4_ cycle genes in the C_4_ species *Setaria viridis* [64]. These findings suggest that while CAM provides an efficient mechanism for adaptation to drought, its efficiency can be significantly impaired under fluctuating or extreme temperature conditions, potentially affecting the plant’s survival and productivity in such environments.

Circadian rhythm plays crucial roles in controlling gene expression and metabolite process in plants, and is involved in protection signaling pathways in response to temperature stress [51,52]. We examined the expression patterns of circadian clock-related genes in response to cold and heat stress (Appendix A) and found several *K. fedtschenkoi* genes were affected under these stress conditions. Specifically, in leaf tissues, the E3 ubiquitin ligase *COP1* was up-regulated at dusk, while *ELONGATED HYPOCOTYL 5* (*HY5*) was down-regulated at dawn and dusk under heat stress conditions (Appendix A and Appendix A). In *Arabidopsis*, increased temperature triggered the activation of *COP1* to repress the stability of its target *HY5* [65]. This suggests a conserved mechanism where COP1-HY5 interaction might be involved in heat stress adaptation in *K. fedtschenkoi*. Further studies are needed to elucidate the interaction and regulation between COP1 and HY5 in response to temperature fluctuations in *K. fedtschenkoi*. In addition, *CCA1* and *RVE8* were down-regulated in response to both low and high temperature stress at dawn. Recent studies showed that *CCA1* and *RVE8* acted as central oscillator regulatory factors involved in *CBF/DREB1*-mediated cold signaling pathway in *Arabidopsis* [52,66]. Under heat stress condition, the transcript abundance of *CCA1* was found to be significantly induced during the daytime (morning and afternoon) according to transcriptomic studies in *Arabidopsis* [51,67]. The differential regulation of *CCA1* and *RVE8* in *K. fedtschenkoi* under cold and heat stress may point to unique adaptations in the circadian clock of CAM plants, revealing potential conservation and divergence from C_3_ species like *Arabidopsis*.

Lastly, we integrated the DEGs identified in our study into the known pathways from C_3_ species that respond to cold and heat stress [13,59,68], highlighting their critical roles in temperature stress signaling in *K. fedtschenkoi* (Figure 7). In particular, under cold stress, the expression of *RVE8* is activated, which in turn up regulates the key transcription factor *CBF4*, leading to the activation of cold-responsive genes such as *COR27* and *COR314-TM2*, enhancing cold tolerance. Conversely, under heat stress, *HSFA2* and *HSFA3* are up-regulated by *HSFA1* and *DERB2A*, initiating downstream signaling pathways to protect against heat stress. The up-regulation of *APX*, *SOD*, and *CAT* genes suggests that high temperatures trigger a robust antioxidant defense mechanism to mitigate oxidative damage.

In conclusion, this study contributes to a deeper understanding of molecular responses to temperature stresses in CAM plants and offers potential targets for genetic engineering to enhance cold and heat stress tolerance in crops, ultimately improving agricultural productivity in the face of climate change. Notably, while CAM species like *K. fedtschenkoi* share some conserved molecular pathways with C_3_ and C_4_ species, such as the regulation of *HSFs*, *HSPs* and antioxidant defense genes (e.g., *APX*, *SOD*, and *CAT*), there are distinct differences in their circadian regulation and other stress response mechanisms. Our findings highlight both shared and divergent molecular strategies employed by CAM plants in adapting to temperature extremes. These insights can inform future attempts to transfer beneficial traits, such as enhanced water-use efficiency and stress resilience, from CAM species into C_3_ crops through genetic engineering, offering promising strategies to increase agricultural resilience in the context of global climate change.

## 4. Materials and Methods

### 4.1. Plant Material and Sample Collection

The fresh *K. fedtschenkoi* (ORNL diploid accession M2) seedlings were propagated from stem cuttings and were grown in a Percival Model AR-75L2 growth chamber for 4 weeks at photon flux density (PFD) of 250 µmol m^−2^ s^−1^ at plant height on a 12 h-light/12 h-dark photoperiod with day/night temperatures of 25 °C/18 °C, respectively. For temperature treatment experiments, plants were then divided into three groups and grown in three separate growth chambers with temperature set at 24 h 37 °C (heat), 8 °C (cold), and 25 °C (control). After 7 days of temperature acclimation, mature leaf (leaf pairs of 4th, 5th, and 6th below the shoot apex) and root tissues were collected at two time points of dawn (2 h before the start of light period) and dusk (2 h before the start of dark period). The samples were then immediately frozen in liquid nitrogen and stored in −80 °C freezer for further RNAs extraction. Three individual plants were used for samples collection at each time point to represent three biological replicates for each treatment.

### 4.2. RNA Extraction

Frozen tissues (leaf or root) were pulverized using a mortar and pestle in liquid nitrogen. Approximately 100 mg of ground tissue was lysed using 850 μL of CTAB buffer (1.0% *β*-mercaptoethanol freshly added) for 5 min at 56 °C with constant mixing on a shaker incubator. Then, 600 μL of chloroform/isoamyl alcohol (24:1) was added and thoroughly mixed. Samples were centrifuged for 8 min at maximum speed, and the resulting supernatant was applied to a filter column. RNA purification steps were then followed as described by Hu et al. [29] using the sigma Spectrum™ Plant Total RNA Kit (Sigma-Aldrich, St. Louis, USA, Cat. No. STRN250-1KT). An on-column DNase treatment was used for elimination of genomic DNA residues, and eluted RNAs were analyzed by a NanoDrop 1000 spectrophotometer (Thermo Scientific, Wilmington, DE, USA) for quantity and purity. A total of 3 µg of RNA for each sample was prepared and shipped to HudsonAlpha Institute for Biotechnology (Huntsville, AL, USA) for library preparation and transcriptomic sequencing.

### 4.3. Library Construction and RNA-Seq

Plate-based RNA sample prep was performed on the PerkinElmer Sciclone NGS robotic liquid handling system using Illumina’s TruSeq Stranded mRNA HT sample prep kit utilizing poly-A selection of mRNA following the protocol outlined by Illumina in their user guide (https://support.illumina.com/sequencing/sequencing_kits/truseq-stranded-mrna.html, accessed on 8 January 2018) and with the following conditions: total RNA starting material was 1 ug per sample, and 8 cycles of PCR were used for library amplification. The prepared libraries were quantified using KAPA Biosystem’s next-generation sequencing library qPCR kit (Roche) and run on a Roche LightCycler 480 real-time PCR instrument. The quantified libraries were then multiplexed, and the pool of libraries was prepared for sequencing on the Illumina HiSeq sequencing platform utilizing a TruSeq paired-end cluster kit, v4, and Illumina’s cBot instrument to generate a clustered flow cell for sequencing. Sequencing of the flow cell was performed on the Illumina HiSeq 2500 sequencer using HiSeq TruSeq SBS sequencing kits, v4, following a 2 × 150 indexed run recipe.

### 4.4. Data Analysis

After filtering out low-quality reads, RNA-Seq reads from each library were aligned to the *K. fedtschenkoi* reference genome using GSNAP (v2018-07-04) [27]. The FeatureCounts function of Subread (v1.6.1) was used to generate raw gene counts, and only reads that mapped uniquely to 1 locus were counted [69]. Gene expression was estimated as transcripts per million (TPM) [70]. DESeq2 (v1.2.10) was subsequently used to determine which genes were differentially expressed between pairs of conditions [71]. The parameters used to “call a gene” between conditions were determined at a false discovery rate (FDR) adjusted *p*-value of ≤0.05. Gene Ontology (GO) enrichment analysis was applied to predict gene function and calculate the functional category using BiNGO [72]. Heat map and bubble plots were generated by the R package ggplot2. For co-expression analysis, the log2-normalized TPM values of all the samples were used to construct a weighted gene co-expression network using the R package WGCNA [73]. All tools were run with default parameters.

## Figures and Tables

**Figure 1 plants-13-03444-f001:**
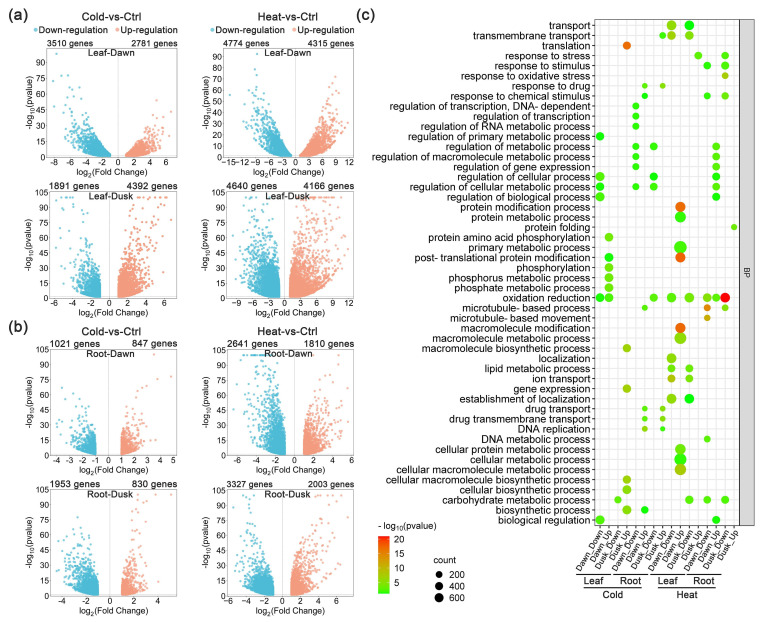
Identification of DEGs regulated by cold or heat stress treatments in *Kalanchoë fedtschenkoi*. (**a**) Volcano plots showing up- and down-regulated genes under cold or heat stress conditions in leaf tissue. (**b**) Volcano plots showing up- and down-regulated genes under cold or heat stress conditions in root tissue. (**c**) GO-enrichment analysis on BP for up- and down-regulated genes at dawn and dusk time points in leaf and root tissues. Detailed enrichment plots on biological process (BP), molecular function (MF), and cellular component (CC) are shown in Appendix A. GO-slim terms are shown here.

**Figure 2 plants-13-03444-f002:**
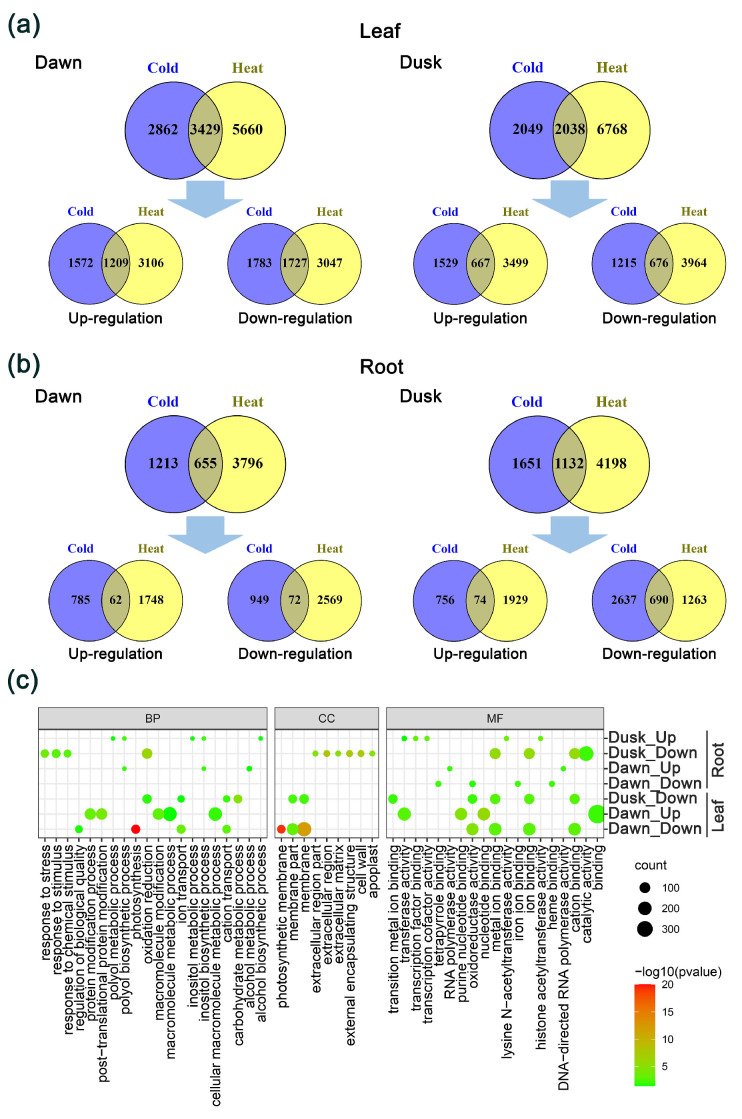
Analysis of DEGs regulated by cold and heat stress in *Kalanchoë fedtschenkoi*. (**a**) Venn diagrams show overlapped DEGs that are regulated by cold and heat stress treatments at dawn and dusk in leaf tissues. (**b**) Venn diagrams show overlapped DEGs that are regulated by cold and heat stress treatments at dawn and dusk in root tissues. (**c**) GO-enrichment analysis for up- and down-regulated genes that are shared between comparisons cold vs. ctrl and heat vs. ctrl at dawn and dusk time points in leaf and root tissues. Detailed enrichment plots on biological process (BP), molecular function (MF), and cellular component (CC) are shown in Appendix A. GO-slim terms are shown here.

**Figure 3 plants-13-03444-f003:**
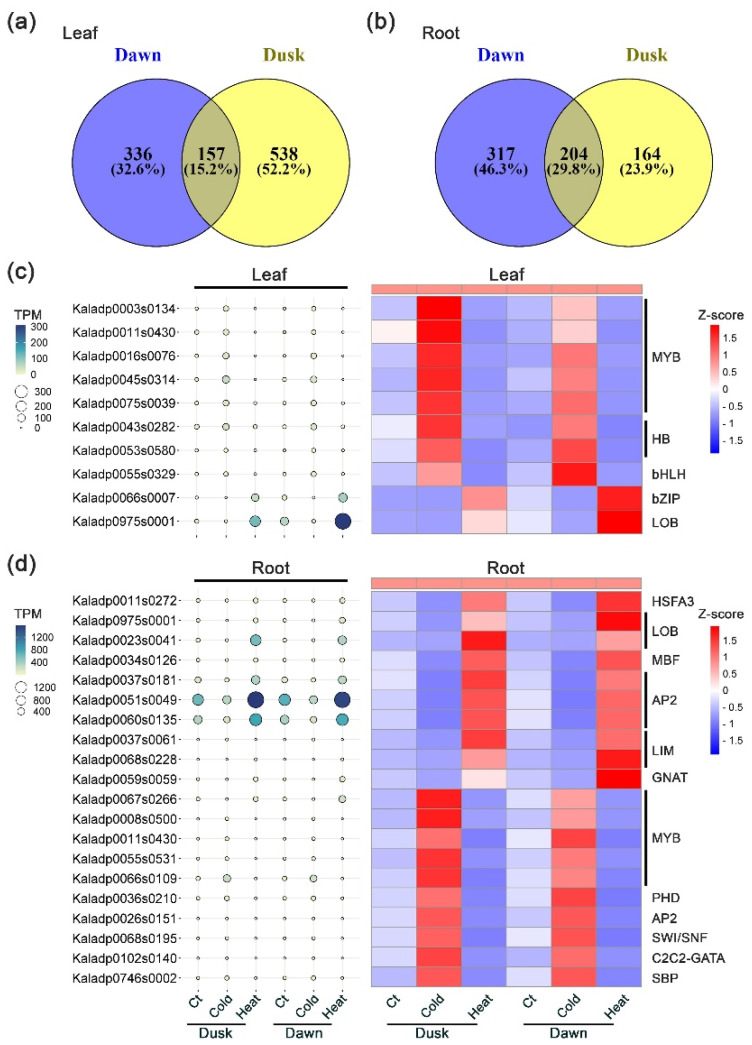
Expression pattern of transcription factors that are oppositely regulated by cold and heat stress in leaf and root tissues at dawn and dusk time points. (**a**) Venn diagrams show overlapped DEGs that are oppositely regulated by cold and heat stress at dawn and dusk in leaf tissues. (**b**) Venn diagrams show overlapped DEGs that are oppositely regulated by cold and heat stress at dawn and dusk in root tissues. (**c**) Expression pattern of TFs that are oppositely regulated by cold and heat stress at dawn and dusk time points in leaf tissues. (**d**) Expression pattern of TFs that are oppositely regulated by cold and heat stress at dawn and dusk time points in root tissues. Z-score was used for normalization of gene transcript profiles and for generation of Heatmap figures.

**Figure 4 plants-13-03444-f004:**
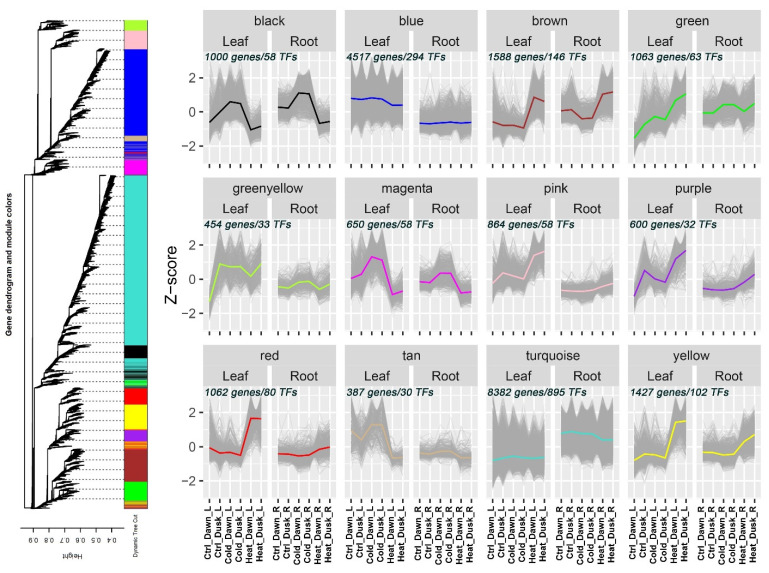
A weighted gene co-expression network analysis (WGCNA) of genes under cold and heat stress treatments in leaf and root tissues of *Kalanchoë fedtschenkoi*. Cluster dendrogram was constructed and different colors were labelled for various expression modules (MEs). Z-score normalization was employed for plot generation.

**Figure 5 plants-13-03444-f005:**
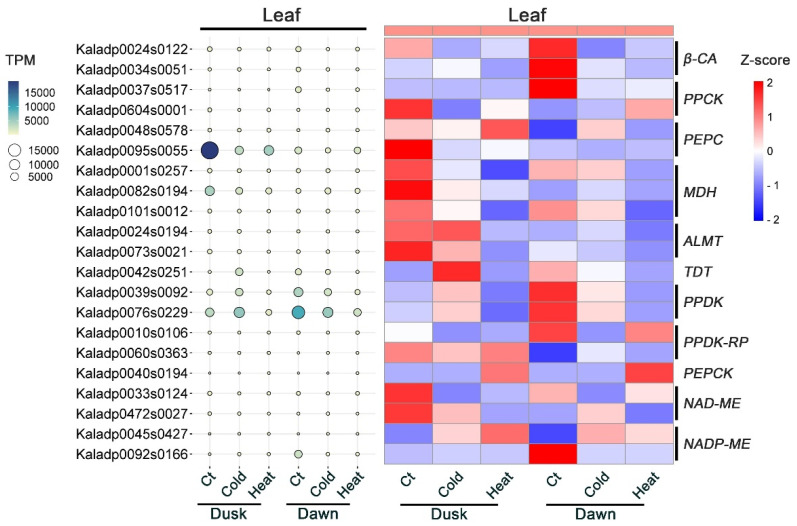
Expression pattern of CAM core genes in response to temperature stresses (cold and heat) at different time points. Z-score was used for normalization of gene transcript profiles and for generation of Heatmap figure. *β-CA*, *β* type carbonic anhydrase; *PEPC*, phosphoenolpyruvate carboxylase; *PPCK*, PEPC kinase; *MDH*, malate dehydrogenase; *ALMT,* tonoplast aluminum-activated malate transporter; *TDT*, tonoplast dicarboxylate transporter; *PPDK*, pyruvate phosphate dikinase; *PPDK-RP*, PPDK regulatory protein; *NAD(P)-ME*, NAD(P)^+^-dependent malic enzyme. Dawn, 2 h before the start of light period; dusk, 2 h before the dark period; Ct, regular growth temperature used as control.

**Figure 6 plants-13-03444-f006:**
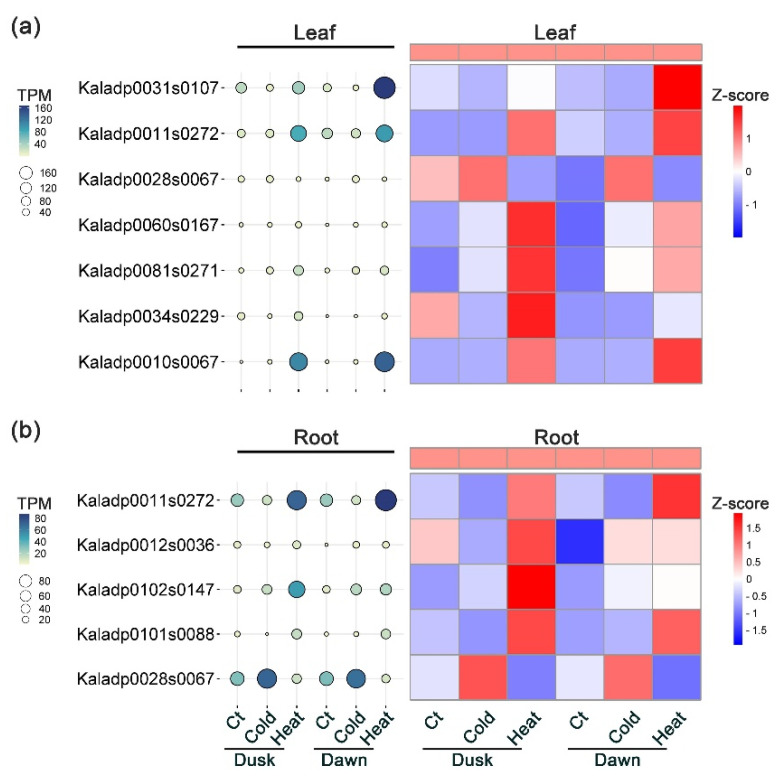
Expression pattern of heat shock factors (HSFs) that are regulated by cold and heat stress in *Kalanchoë fedtschenkoi*. (**a**) Transcript profiles of HSFs regulated by both cold and heat stress in leaf tissues. (**b**) Transcript profiles of HSFs regulated by both cold and heat stress in root tissues. Z-score was used for normalization of gene transcript profiles and for generation of Heatmap figure.

**Figure 7 plants-13-03444-f007:**
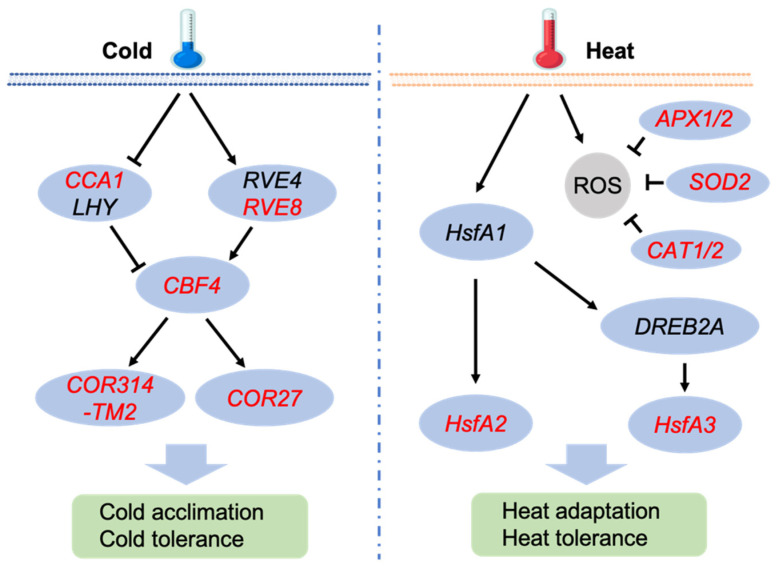
A schematic diagram showing pathways responsive to cold and heat stress in plants. Red color indicates the genes identified in this study. CCA1, Circadian Clock Associated 1; LHY, late elongated hypocotyl; RVE4/8, Reveille 4/8; COR314-TM2, cold-regulated 314 thylakoid membrane 2; APX1/2, ascorbate peroxidase 1/2; SOD2, superoxide dismutase 2; CAT1/2, catalase 1/2; HsfA1/A2/A3, heat shock factor A1/A2/A3; DREB2A, dehydration-responsive element binding protein 2; ROS, reactive oxygen species.

**Table 1 plants-13-03444-t001:** DEGs responsible for cold and heat stress in *Kalanchoë fedtschenkoi.*

Comparison				DEGs		
Type	Tissue	Time	Treatment	Total	Up-Regulated	Down-Regulated
Temperature	Leaf	Dawn	Cold vs. Ctrl	6291	2781	3510
			Heat vs. Ctrl	9089	4315	4774
		Dusk	Cold vs. Ctrl	4087	2196	1891
			Heat vs. Ctrl	8806	4166	4640
	Root	Dawn	Cold vs. Ctrl	1868	847	1021
			Heat vs. Ctrl	4451	1810	2641
		Dusk	Cold vs. Ctrl	2783	830	1953
			Heat vs. Ctrl	5330	2003	3327

## Data Availability

All raw short reads are available in the NCBI SRA database (SRA accessions: SRX4497844-SRX4497879).

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
