# Peer review of "Transcriptomic Analysis of the CAM Species Kalanchoë fedtschenkoi Under Low- and High-Temperature Regimesâ€"

_plants, 2024, doi:10.3390/plants13233444_

Round 1
Reviewer 1 Report
Comments and Suggestions for Authors
Climate change is a reality, and global temperature rise is leading to extreme temperatures throughout the world. The article "Gene expression regulation in the CAM plant Kalanchoe fedtschenkoi in response to low and high temperatures" shed light on the transcriptomic response of an understudied CAM plant. The paper exhibits a strong writing style and offers a unique perspective. However, I would recommend making minor revisions before publication, as detailed in the comments below:
General Comments
Most of the time, the authors used very old citations, which can be changed with recent citations.
It would be better if the author could also provide the phenotypic pictures of the plants under heat and cold stress.
You can convert Table 2 into bar graphs to improve the visualization and comprehension of the results.
Title
The title should be changed to: Transcriptomic analysis of CAM plant Kalancho fedtschenkoi under low and high temperature regimes
Keywords
Change the keywords that aren't in the paper's title to boost readership and database searches.
Introduction
Lines 48-50 exhibit an excessive number of references. I understand that the author has referenced numerous crops, but given that these are general lines, the author could potentially rephrase them and incorporate only one or two of the most recent citations from 2023-2024.
I would suggest adding some lines about Kalancho Fedtschenkoi studies with abiotic stressors in the introduction, if they are available.
Results
Table 1: The column size of the type can be increased so it can easily accommodate temperature words.
Figure 2: The Venn diagrams are tiny and difficult to read. I would suggest increasing the size and their text for a better reading experience.
Author Response
Climate change is a reality, and global temperature rise is leading to extreme temperatures throughout the world. The article "Gene expression regulation in the CAM plant Kalanchoe fedtschenkoi in response to low and high temperatures" shed light on the transcriptomic response of an understudied CAM plant. The paper exhibits a strong writing style and offers a unique perspective. However, I would recommend making minor revisions before publication, as detailed in the comments below:
General Comments
Most of the time, the authors used very old citations, which can be changed with recent citations.
[Response] Thank you for your suggestions. We have updated the reference list and added several recent publications.
It would be better if the author could also provide the phenotypic pictures of the plants under heat and cold stress.
[Response] Thank you for your suggestions. Unfortunately, we currently do not have pictures of the Kalanchoë fedtschenkoi plants under cold and heat stress conditions. Since the project supporting the experimental work for this manuscript has ended, there are no resources to generate additional data in our lab.
You can convert Table 2 into bar graphs to improve the visualization and comprehension of the results.
[Response] Thank you for your suggestions. There is no Table 2 in the main text. Instead, we provided volcano plots (Fig. 1a-b) to visualize the results presented in Table 1.
Title
The title should be changed to: Transcriptomic analysis of CAM plant Kalancho fedtschenkoi under low and high temperature regimes
[Response] Thank you for your suggestion. We have revised the title to: ‘Transcriptomic Analysis of the CAM Species Kalanchoë fedtschenkoi Under Low and High Temperature Regimes’.
Keywords
Change the keywords that aren't in the paper's title to boost readership and database searches.
[Response] Thank you for your suggestions. We have updated the keywords to “Crassulacean acid metabolism, RNA-Seq, temperature stress, gene regulation, heat shock proteins”.
Introduction
Lines 48-50 exhibit an excessive number of references. I understand that the author has referenced numerous crops, but given that these are general lines, the author could potentially rephrase them and incorporate only one or two of the most recent citations from 2023-2024.
[Response] Thank you for your suggestions. We have revised the citations and included only a select few recent references (see lines 47 - 48 in the revised manuscript).
I would suggest adding some lines about Kalanchoe Fedtschenkoi studies with abiotic stressors in the introduction, if they are available.
[Response] Thank you for your suggestions. We have added more information regarding recent studies on abiotic stressors in the introduction section (see lines 83 - 86 in the revised manuscript).
Results
Table 1: The column size of the type can be increased so it can easily accommodate temperature words.
[Response] Thank you for your comments. We have revised Table 1 accordingly.
Figure 2: The Venn diagrams are tiny and difficult to read. I would suggest increasing the size and their text for a better reading experience.
[Response] Thank you for your comments. We have revised the Venn diagrams in Figure 2 and enlarged the text for improved visualization.
Reviewer 2 Report
Comments and Suggestions for Authors
The authors of the paper to determine the effect of hot and cold temperature exposure to the CAM plant Kalanchoe fedtschenkoi. The goal of this manuscript was to determine how CAM plants compare in their response to high and low temperature to C3 and C4 plants. They exposed plants for one week to the experimental temperatures and looked at genes in RNA in relation to control plants. The authors used observed both leaf and root tissues for differential gene expression. The results found many genes up and down regulated compared to the control plants. The paper describes the concordance of the genes between leaf and root tissue for gene expression. It finishes with looking at CAM related gene expression and found changes compared to the control plants. The discussion compares the results to C3 and C4 plants and in addition to peroxidase genes, transporters and metabolite genes. They found conserved gene expression between C3 and CAM plants.
The experiment was designed well and the paper presents interesting results. The results are presented well but at some points they present broad results mixed with very specific genes making some of the paragraphs extremely dense to read. This makes it more difficult to compare treatments and the differences in leaf and root tissue. The authors present interesting results on heat shock factors and proteins. The section on the expression CAM and circadian rhythm related genes should be a more prominent in the paper since the authors are focusing on CAM plants. The conclusions are sound and do add to the knowledge cold and hot treatment effects on gene regulation in CAM plants. The conclusion is broad about genetic engineering targets as more scientific data would be needed to understand which target genes would be beneficial.
Comments:
The introduction would benefit from more about the life history of Kalanchoe fedtschenkoi. Such as where it grows, light environment, climate as such. Not all CAM plants grow in the desert and Kalanchoe is native to Madagascar and not a desert like Opuntia. There are also C4 plants adapted to desert environments. Most CAM plants are tropical and found in the Bromeliads and Orchids.
Line 72—Malate is produced via carboxylation and forms malic acid in the vacuole lower pH.
The authors use the designation of samples taken at dawn and dusk but samples are collected 2 hours prior to those points. This is not really dawn and dusk but phase I and phase IV or pre-dawn and pre-dusk. This fits more into the circadian rhythms of CAM species.
Fig 1 and Fig 2 – The fonts of the graphs are small and hard to read in paper format.
The authors do not present any data on the relative CAM activity of the control, hot and cold treatments. It would have been helpful to determine AM and PM acid levels to determine relative CAM activity. In addition, since all three treatments held constant temperatures, it is hard to determine the changes in CAM activity. Research has shown day/night temperature differentials are more conducive to CAM as stated in the introduction. It would have been interesting to see the control CAM gene expression with the original temperatures of 25/18 C.
This paper presents a good foundation for the study of gene expression in CAM plants and more studies can be based on this research.
Author Response
The authors of the paper to determine the effect of hot and cold temperature exposure to the CAM plant Kalanchoe fedtschenkoi. The goal of this manuscript was to determine how CAM plants compare in their response to high and low temperature to C3 and C4 plants. They exposed plants for one week to the experimental temperatures and looked at genes in RNA in relation to control plants. The authors used observed both leaf and root tissues for differential gene expression. The results found many genes up and down regulated compared to the control plants. The paper describes the concordance of the genes between leaf and root tissue for gene expression. It finishes with looking at CAM related gene expression and found changes compared to the control plants. The discussion compares the results to C3 and C4 plants and in addition to peroxidase genes, transporters and metabolite genes. They found conserved gene expression between C3 and CAM plants.
The experiment was designed well and the paper presents interesting results. The results are presented well but at some points they present broad results mixed with very specific genes making some of the paragraphs extremely dense to read. This makes it more difficult to compare treatments and the differences in leaf and root tissue. The authors present interesting results on heat shock factors and proteins. The section on the expression CAM and circadian rhythm related genes should be a more prominent in the paper since the authors are focusing on CAM plants. The conclusions are sound and do add to the knowledge cold and hot treatment effects on gene regulation in CAM plants. The conclusion is broad about genetic engineering targets as more scientific data would be needed to understand which target genes would be beneficial.
Comments:
The introduction would benefit from more about the life history of Kalanchoe fedtschenkoi. Such as where it grows, light environment, climate as such. Not all CAM plants grow in the desert and Kalanchoe is native to Madagascar and not a desert like Opuntia. There are also C4 plants adapted to desert environments. Most CAM plants are tropical and found in the Bromeliads and Orchids.
[Response] Thank you for your suggestions. We have incorporated additional information about the life history of Kalanchoë fedtschenkoi (Please see lines 75 – 78 in the revised manuscript).
Line 72—Malate is produced via carboxylation and forms malic acid in the vacuole lower pH.
[Response] Thank you for your comments. We have revised that section to read: ‘into malate, which is then converted to malic acid and stored in the vacuole.’ (Please see lines 71 – 72 in the revised manuscript)
The authors use the designation of samples taken at dawn and dusk but samples are collected 2 hours prior to those points. This is not really dawn and dusk but phase I and phase IV or pre-dawn and pre-dusk. This fits more into the circadian rhythms of CAM species.
[Response] Thank you for your comments. We agree that phase I and phase IV align more closely with the circadian rhythms. However, we have used the terms "dawn" and "dusk" to describe the sampling times in our previous studies (Yang et al., 2018; Zhang et al., 2020; Hu et al., 2022). To be consistent with our previous publications, it would be better to keep these designations in this manuscript.
Fig 1 and Fig 2 – The fonts of the graphs are small and hard to read in paper format.
[Response] As suggested, we have increased the fonts in all the figures.
The authors do not present any data on the relative CAM activity of the control, hot and cold treatments. It would have been helpful to determine AM and PM acid levels to determine relative CAM activity. In addition, since all three treatments held constant temperatures, it is hard to determine the changes in CAM activity. Research has shown day/night temperature differentials are more conducive to CAM as stated in the introduction. It would have been interesting to see the control CAM gene expression with the original temperatures of 25/18 C.
[Response] Thank you for your suggestions. Since the project supporting the experimental work for this manuscript has ended, there are no resources to generate the CAM activity data in our lab. We will pursue new funding opportunities for gathering such data, aiming to deepen our understanding of molecular regulation in CAM species in response to environmental stimuli.
This paper presents a good foundation for the study of gene expression in CAM plants and more studies can be based on this research.
[Response] We sincerely appreciate your kind support for our study.